# Breaking the nanoparticle's dispersible limit via rotatable surface ligands

Yue Liu[1], Na Peng[2,3], Yifeng Yao[4], Xuan Zhang [4], Xianqi Peng[2], Liyan Zhao[1], Jing Wang [5], Liang Peng[6], Zuankai Wang [6], Kenji Mochizuki [4✉], Min Yue [2,3,7✉] & Shikuan Yang [1,7,8,9✉]

Achieving versatile dispersion of nanoparticles in a broad range of solvents (e.g., water, oil, and biofluids) without repeatedly recourse to chemical modifications are desirable in optoelectronic devices, self-assembly, sensing, and biomedical fields. However, such a target is limited by the strategies used to decorate nanoparticle's surface properties, leading to a narrow range of solvents for existing nanoparticles. Here we report a concept to break the nanoparticle's dispersible limit via electrochemically anchoring surface ligands capable of sensing the surrounding liquid medium and rotating to adapt to it, immediately forming stable dispersions in a wide range of solvents (polar and nonpolar, biofluids, etc.). Moreover, the smart nanoparticles can be continuously electrodeposited in the electrolyte, overcoming the electrode surface-confined low throughput limitation of conventional electrodeposition methods. The anomalous dispersive property of the smart Ag nanoparticles enables them to resist bacteria secreted species-induced aggregation and the structural similarity of the surface ligands to that of the bacterial membrane assists them to enter the bacteria, leading to high antibacterial activity. The simple but massive fabrication process and the enhanced dispersion properties offer great application opportunities to the smart nanoparticles in diverse fields.

[1] Institute for Composites Science Innovation, School of Materials Science and Engineering, Zhejiang University, Hangzhou 310027, China. [2] Institute of Veterinary Sciences & Department of Veterinary Medicine, College of Animal Sciences, Zhejiang University, Hangzhou 310058, China. [3] Hainan Institute of Zhejiang University, Sanya 572025, China. [4] Department of Chemistry, Zhejiang University, Hangzhou 310028, China. [5] Department of Mechanical Engineering, University of Michigan, Ann Arbor, MI 48109, USA. [6] Deparment of Mechanical Engineering, City University of Hongkong, Hongkong 999077, China. [7] Department of Medical Oncology, The first affiliated Hospital, Zhejiang University School of Medicine, Hangzhou 310003, China. [8] State Key Laboratory of Fluid Power and Mechatronic Systems, Zhejiang University, Hangzhou 310027, China. [9] Baotou Research Institute of Rare Earths, Baotou 014030, China. ✉email: kenji_mochizuki@zju.edu.cn; myue@zju.edu.cn; shkyang@zju.edu.cn

Colloidal solutions composed of dispersed nanoparticles in a continuous liquid medium are important for many applications[1–4]. The surface of the nanoparticles is usually capped by macromolecules, other particles, or ions to increase their affinity to the solvent and to strengthen the inter-particle repulsion force to form stable colloids[5,6]. Existing nanoparticles can only disperse in a narrow range of solvents (i.e., dispersible limit) and specific chemical modifications have to be performed to disperse the nanoparticles in specific solvents[7,8]. Concepts that can expand the solvent types that the nanoparticles can disperse in (or breaking the dispersible limit) are highly desirable for simplifying the practical application processes and offering new application opportunities. Although many approaches have been explored (e.g., designing the structure of the surface ligands or using multiple surface ligands), the nanoparticle's dispersible solvent types have only been slightly expanded. Here we show a concept to break the nanoparticle's dispersible limit via rotatable surface ligands.

Electrostatic repulsive forces between charged nanoparticles are crucial to forming stable colloids in polar solvents (Scheme I in Fig. 1a), while the electric double-layer surrounding the nanoparticles collapses in nonpolar solvents and flocculation immediately takes place. Hydrocarbon chains (or surface ligands) tethered to the nanoparticles can increase the inter-nanoparticle repelling force in nonpolar solvents to form sterically stabilized

nanoparticle colloids (Scheme II in Fig. 1a), while the ligand chains tend to contract and in turn results in the formation of nanoparticle aggregations in polar solvents[5,7]. Therefore, nanoparticles grafted with a specific type of surface ligands can only disperse in a narrow range of solvents or a dispersible limit exists for specific nanoparticles[7] (Fig. 1b). A compromising way to expand the nanoparticle's dispersible solvent types is through simultaneously anchoring different surface ligands or complex structured ligands with varied properties at different parts (e.g., amphiphilic ligands) onto the nanoparticles (Scheme III and IV in Fig. 1a)[8,9]. However, this compromising way can only slightly expand the solvent types because of the limited conformational structure change degree of the ligands[10,11] (Fig. 1b). Here we introduce rotatable surface ligands (Scheme V in Fig. 1c) to break the nanoparticle's dispersible limit: the electrochemically anchored surfactant ligands via non-directional ionic bonds to the nanoparticle surface can sense the surrounding liquid medium and rotate to adapt to it, providing an effective avenue towards creating smart nanoparticles dispersible in a wide range of solvents [e.g., from polar water (polarity: 10.2) to nonpolar cyclohexane (polarity: 0.1)] (Scheme V in Fig. 1b). The smart nanoparticles can avoid aggregation even exposing to bacteria secreted species and the surface ligands help the smart nanoparticle to penetrate the bacterial membrane, showing high antibacterial performance (Fig. 1d).

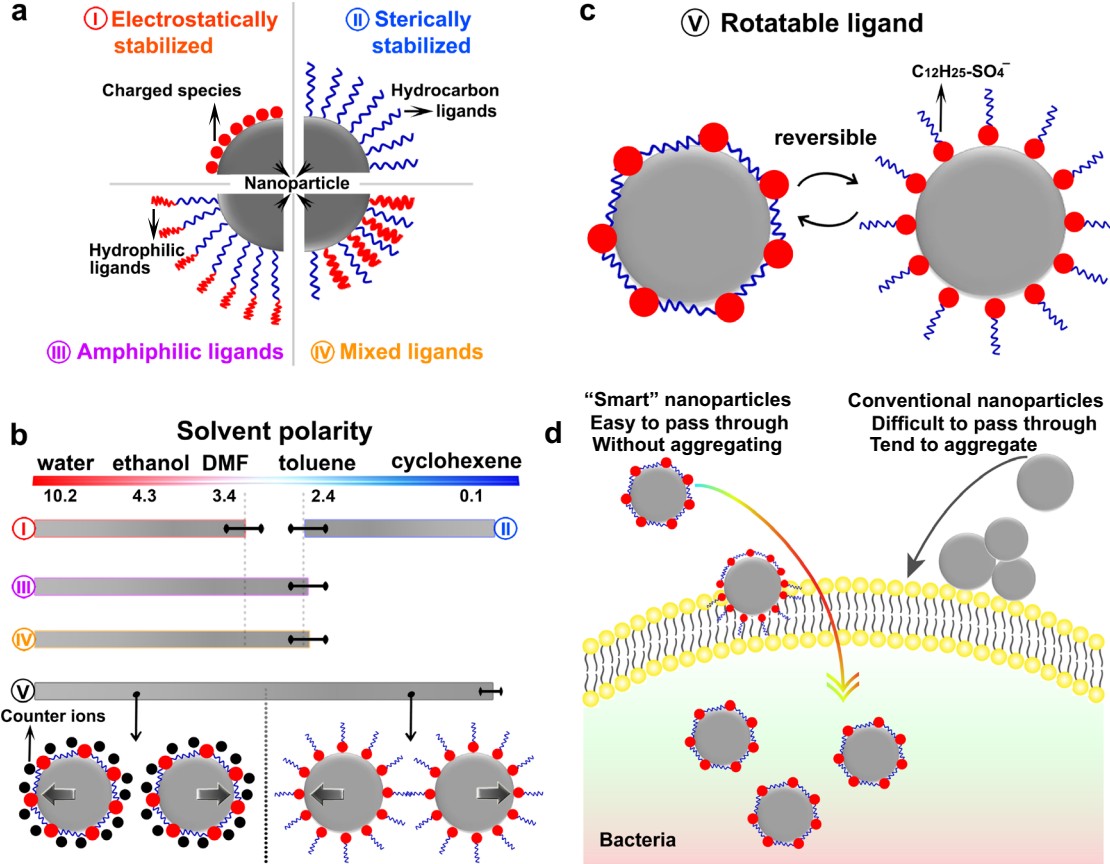

**Fig. 1 Existence of a dispersible limit for specific nanoparticles. a** Different strategies used to stabilize nanoparticles in different solvents. (I) Electrostatically stabilized nanoparticles in polar solvents. (II) Sterically stabilized nanoparticles in nonpolar solvents. (III) Amphiphilic ligands stabilized nanoparticles. (IV) Mixed ligands stabilized nanoparticles. **b** The appropriate dispersible limit of different stabilizing strategies shown in (**a**) and breaking the dispersible limit of common stabilizing strategies by tethering rotatable surface ligands onto nanoparticles (V). The rotatable surface ligands allow the nanoparticles to be electrostatically stabilized in polar solvents, while sterically stabilized in nonpolar solvents. **c** Reversible orientation change of the rotatable ligands to adapt to surrounding liquids. **d** The high antibacterial performance of the smart Ag nanoparticles arises from the fact that they do not aggregate and they can easily pass through the bacterial membrane.

## Results

**Continuous fabrication and formation mechanism of smart Ag nanoparticles**. The smart Ag nanoparticles were electrodeposited in an electrolyte solution composed of silver nitrate (AgNO₃) and sodium dodecyl sulfate ($C_{12}H_{25}SO_4Na$ or SDS). Different from conventional electrodeposition that exclusively confined electrodeposited nanostructures on the electrode surface[12–18], here the Ag nanoparticles continuously jetted into the electrolyte solution from the electrode surface, forming yellow mist-like trajectories (Fig. 2a and Supplementary Movie 1). Yellow colloidal solutions were obtained within 20 s (Supplementary Fig. 1), manifested by the prominent Tyndall effect[19] (inset in Fig. 2b). An absorption peak located at 418 nm was observed from the colloidal solution, originating from the localized surface plasmon resonance (LSPR) of the Ag nanoparticles[20] (Fig. 2b). Multiple characterization methods have been used to confirm the formation of Ag nanoparticles (Supplementary Fig. 2). The nanoparticles formed under 30 V are quasi-spherical with a mean size of ~25 nm (Fig. 2c). The mean size of the Ag nanoparticles gradually decreased from ~50 to ~12 nm as the deposition voltage was increased from 10 to 100 V (Fig. 2d and Supplementary Fig. 3). Meanwhile, the nanoparticles tended to be more spherical at high deposition voltages (Supplementary Fig. 4).

Overcoming the constraint of the electrode surface during electrodeposition endows the capability to continuously produce nanoparticles in the electrolyte solution. The existence of SDS played significant roles in producing Ag nanoparticles in the electrolyte solution, otherwise, Ag nanoparticles were formed exclusively on the electrode surface (Supplementary Movie 2). As the amount of SDS was increased in the electrolyte solution, more Ag nanoparticles started to spread from the electrode surface to the electrolyte solution. After electrodeposition at 30 V for 5 min, we calculated the production rate of the Ag nanoparticles by weighing their weight at different SDS concentrations (Supplementary Fig. 5). As the concentration of SDS was increased from 3 mM to 10 mM, and further to 90 mM, the yield rate of the Ag nanoparticles was increased from ~70 μg/s, to ~160 μg/s, and further to ~260 μg/s using 30 ml of the electrolyte composed of 30 mM AgNO₃ (Fig. 2e). This means that SDS can efficiently promote the formation of Ag nanoparticles in the electrolyte. The physical properties of the electrode could also influence the production efficiency, which was increased by ~10 times to ~2000 μg/s after replacing the smooth aluminum foil electrode with a cylindrical rough carbon rod (Supplementary Fig. 6). Notably, the production rate of the Ag nanoparticles using our electrodeposition method is even higher than the wet chemical methods (~50 μg/s) (Supplementary Fig. 6). After removing the Ag nanoparticles from the electrolyte after 30 V electrodeposition for 300 s by centrifugation, no Ag nanoparticles could be electrochemically formed because all of the silver ions had been consumed (Supplementary Fig. 7).

It is a prerequisite to figure out the formation mechanism of the Ag nanoparticles in the electrolyte solution to achieve the shape, size, and production efficiency control over the Ag nanoparticles. Dodecyl sulfate ions ($C_{12}H_{25}SO_4^-$) have strong interactions with silver ions in the electrolyte solution, manifested by the appearance of the 961 cm⁻¹ Raman peak associated with the silver dodecyl sulfate[21,22] (Supplementary Fig. 8). When the silver ions were reduced on the cathode electrode surface to form small Ag nanoparticles, dodecyl sulfate ions were anchored onto the Ag nanoparticles (Process I in Fig. 2f). The dodecyl sulfate ligands make the Ag nanoparticles negatively charged (Process II in Fig. 2f), verified by the Zeta potential measurement (~ −30 mV). The shear force $f_1$ caused by the electrodynamic force and the bubble formation/release from the electrode surface

works with the electrostatic repulsive force $f_2$ between the cathode electrode and the negatively charged silver nanoparticles to detach the nanoparticles from the cathode electrode surface (Process III in Fig. 2f, g). The growth time of the Ag nanoparticles equals their connection time to the electrode surface, which determines their eventual size and is dependent on the competition between the adhesive force $f_3$ at the interface of the Ag nanoparticles and the electrode and the ripping force of $f_1$ and $f_2$ (Fig. 2g). When the deposition voltage was increased, the shear force $f_1$ was strengthened, giving rise to the formation of smaller Ag nanoparticles (Supplementary Fig. 3). More dodecyl sulfate ions were anchored onto the Ag nanoparticles when the concentration of SDS was increased, making them more negatively charged, which in turn enhanced $f_2$. Therefore, the size of the Ag nanoparticles was slightly reduced at higher SDS concentrations (Supplementary Fig. 9). When the aluminum foil cathode electrode was replaced by a rough carbon rod, the adhesive force $f_3$ between the Ag nanoparticle and the electrode was significantly weakened. Therefore, the Ag nanoparticles could be easily peeled off from the carbon rod, leading to a substantially increased production rate (Supplementary Fig. 6).

The Ag nanoparticles can be continuously prepared by constructing an automatic apparatus mainly composed of an electrochemical cell and a filtration setup (Supplementary Fig. 10). A water pump was used to transport the electrodeposited colloidal solutions into the filtration system equipped with an ultrafiltration membrane (pore size: ~10 nm). Ag nanoparticles were continuously collected on the filtration membrane (Fig. 2h). The filtrated solution without Ag nanoparticles was pumped back into the electrochemical cell for further usage. In principle, the yield rate of the Ag nanoparticles can reach ~7 g/h using the automatic apparatus given that AgNO₃ is adequately replenished into the electrolyte solution (30 ml). We envision that the automatic apparatus's yield rate of the Ag nanoparticles can be increased to be kg/h by using multiple thick cathode carbon rods and more electrolyte solutions.

Inspired by the 3D printing technique[23], an electrochemical pen was constructed capable of writing conductive circuits formed by smart Ag nanoparticles on arbitrary substrates (Fig. 2i). The cathode wire was fixed closely to the anode needle and the two could be connected by an electrolyte droplet hanging on the needle tip generated by pushing the syringe. Ag nanoparticles were immediately formed within the electrolyte droplet. Slowly moving and pushing the syringe allowed us to print smart Ag nanoparticles into electric circuits onto glass, plastic, ceramics, silicon, metal mesh, and flexible polydimethylsiloxane (PDMS) surfaces (Fig. 2j). The broken circuits can be conveniently remediated by the electrochemical pen (Fig. 2k).

Polyvinyl pyrrolidone (PVP) has been widely used to shape Ag nanoparticles during wet chemical fabrication processes[24], because it can attach onto the Ag nanoparticle surface. We replaced SDS with PVP in the electrolyte solution and a dark yellow mist was appeared when 30 V was applied (Supplementary Fig. 11). However, the yellow colloidal solution was not stable and the Ag nanoparticles precipitated at the bottom after 7 days. In contrast, the colloidal solution prepared with SDS was very stable and without observable precipitations even after preserving for 6 months (Supplementary Fig. 12). This encouraged us to study the dispersive properties of the electrodeposited Ag nanoparticles prepared with SDS.

**Anomalous dispersive properties of smart Ag nanoparticles**. Different from common nanoparticles, the electrodeposited Ag nanoparticles demonstrated anomalous dispersive properties, immediately forming stable colloidal dispersions in water

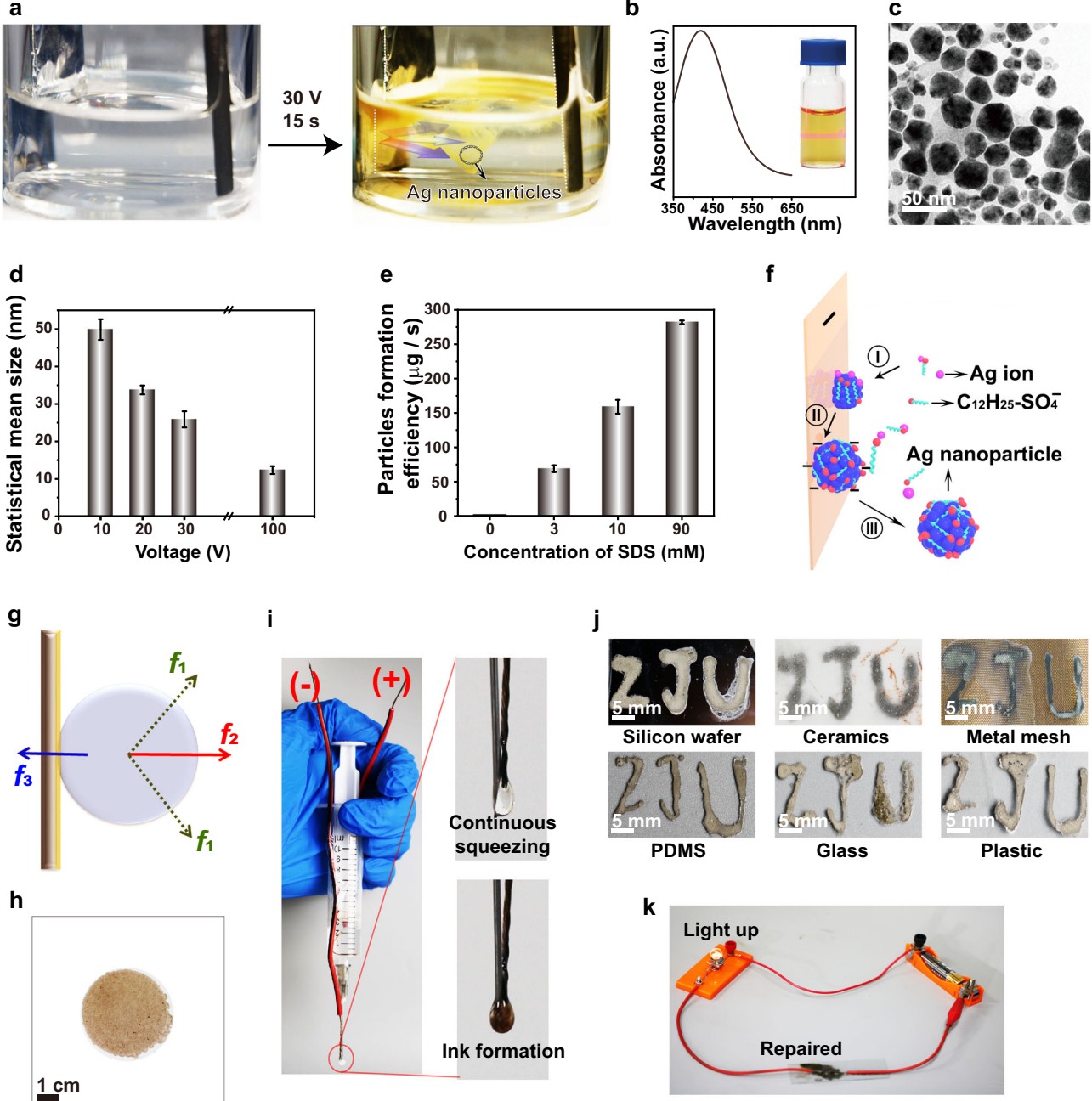

**Fig. 2 Continuous electrochemical fabrication and characterization of the smart Ag nanoparticles. a** Smart Ag nanoparticles were continuously formed at 30 V in aqueous solutions. The photos were captured from Supplementary Movie 1. **b** The absorption spectrum of the Ag nanoparticle colloid in water. Inset: Tyndall effect was observed when a laser beam passing through the colloidal solution. **c** Transmission electron microscope image of the prepared Ag nanoparticles at 30 V for 30 s. **d** The mean size of the Ag nanoparticles fabricated at different voltages. **e** The production efficiency of Ag nanoparticles in electrolytes with different SDS concentrations at 30 V. **f** Schematic of the formation process of the Ag nanoparticles. Process I: the nuclei are formed on the electrode surface passivated by dodecyl sulfate ions. Process II: The nanoparticles are negatively charged induced by the dodecyl sulfate ligands. Process III: The Ag nanoparticles are detached from the electrode surface. **g** Schematic of the forces exerted to the Ag nanoparticles on the electrode surface. $f_1$, $f_2$, and $f_3$ represent the shear force induced by the electrodynamic force and the formation/release of hydrogen bubbles, the electrostatic repulsive force between the nanoparticle and the electrode, and the adhesive force between the Ag nanoparticle and the electrode, respectively. **h** A large amount of Ag nanoparticles formed on the filtration membrane. **i** The electrochemical pen. Inset: an electrolyte droplet is formed by pushing the syringe, which can connect the anode and the cathode wires to generate Ag nanoparticles colloidal ink. **j** The ZJU letters can be written by the electrochemical pen on different substrates. **k** Repairing the broken circuit by the electrochemical pen. Error bars in (**d**, **e**) represent standard deviation of five independent measurements.

(polarity: 10.2), common organic liquids [e.g., glycol (polarity: 6.9), DMF (polarity: 3.4), toluene (polarity: 2.4), and cyclohexane (polarity: 0.1)], and even biofluids (e.g., plasma) (Fig. 3a and Supplementary Fig. 13). No precipitates were observed after at least 7 days in the above liquids (Supplementary Fig. 13). No

aggregation formed even after heating the aqueous colloids at 90 °C for 24 h and after freezing and thawing the colloids (Supplementary Fig. 14). The colloidal solutions kept stable even in the presence of <0.5 M of NaNO$_3$ (Supplementary Fig. 14). The LSPR peak of the Ag nanoparticles located at around 420 nm and

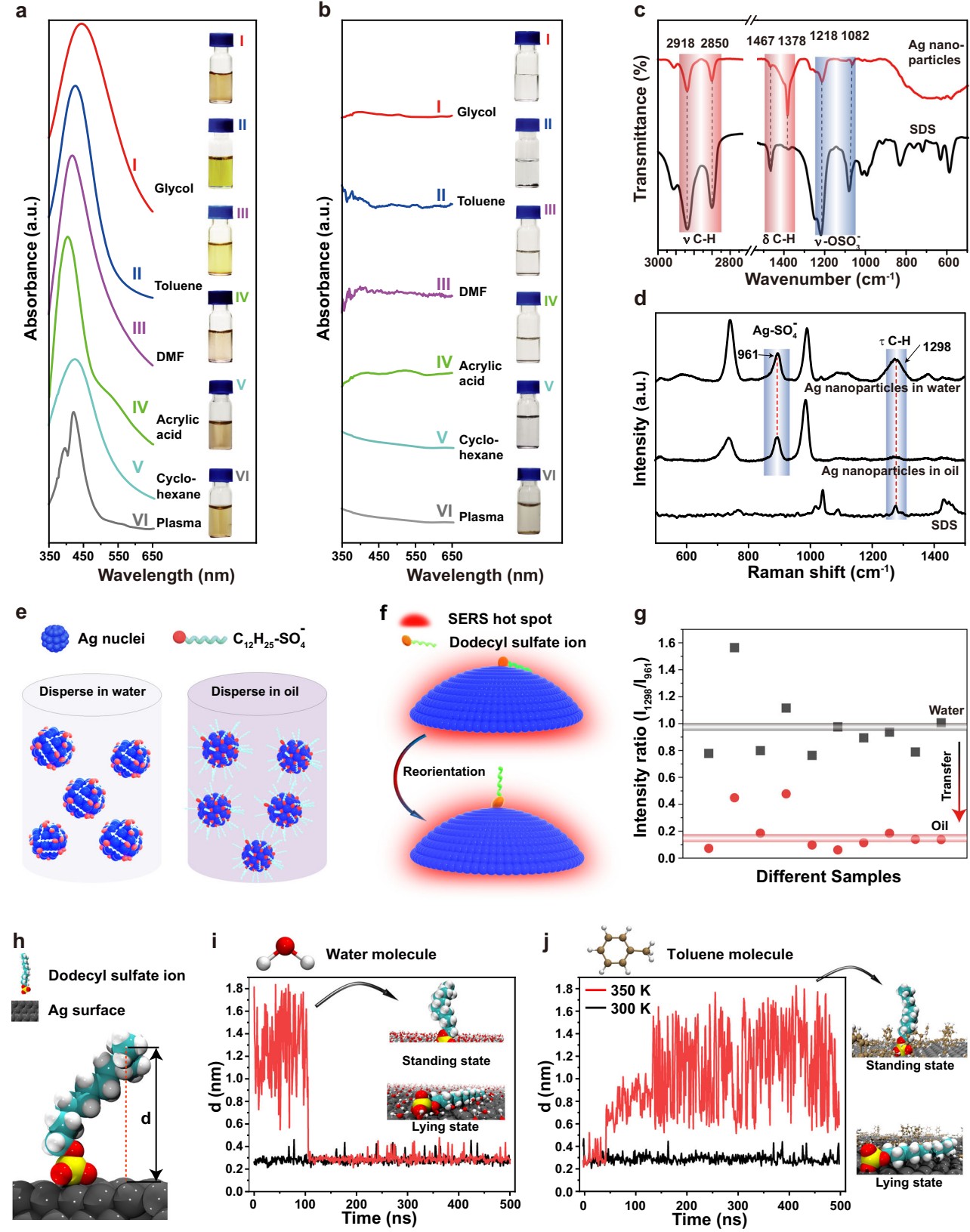

shifted slightly in different liquids, because of their different refractive index[25,26].

Comparatively, Ag nanoparticles prepared by the wet chemical method could only disperse in water (Supplementary Fig. 15), but not organic liquids and plasma (Fig. 3b). Au nanoparticles grafted with the cationic surfactant n-hexadecyltrimethylammonium ions could only disperse in polar solvents (e.g., water, DMF, and ethanol), while immediately aggregated once met less and nonpolar solvents (e.g., toluene and cyclohexane)[27,28] (Supplementary Fig. 16).

**Fig. 3 Rotatable surface ligands endowed the smart Ag nanoparticles with anomalous dispersive properties. a** The absorption spectra of the smart Ag nanoparticles dispersed in different organic liquids without any chemical modification after electrodeposition in aqueous solution. **b** Ag nanoparticle prepared by wet chemical method in water immediately aggregated in organic liquids. Insets in (**a**), (**b**) are photos of the dispersive status of the Ag nanoparticles in different liquids. **c** FTIR spectra of the smart Ag nanoparticles and SDS powder. **d** SERS spectra of the smart Ag nanoparticles dispersed in water and oil and the SDS powder. **e** Smart Ag nanoparticles exposing hydrophilic sulfate head in water and hydrophobic alkyl chain in oil. **f** Schematic of the orientation change of the dodecyl sulfate surface ligands when transfer the smart Ag nanoparticle from water to oil, as well as the mechanism to monitor the orientation change by SERS. **g** The evolution of the intensity ratio between the 1298 cm$^{-1}$ (from the dodecyl tail) and the 961 cm$^{-1}$ (from sulfate head) SERS peaks when transfer the smart Ag nanoparticles from water to oil (e.g., toluene). The data were collected from 10 different samples. **h** The vertical distance between the end carbon atom in the dodecyl chain and the Ag surface was defined as *d*. **i** MD simulation of the *d* variation when the ligand at the lying state (black curve) and the standing state (red curve) exposed to water. Inset: the ligand at the standing state and the lying state in water captured from Supplementary Movie 3. **j** MD simulation of the *d* variation when the ligand at lying state exposed to toluene at 300 K (black curve) and 350 K (red curve). Inset: the ligand at the standing state and the lying state in toluene captured from Supplementary Movie 4. Note: In the insets, only the solvent molecules on the Ag surface were shown, while the ligand was fully immersed in the solvent in MD simulations.

**Physical origin of the anomalous dispersive properties**. Fourier transform infrared spectroscopy (FTIR) characterization indicated that dodecyl sulfate ions coexisted with the Ag nanoparticles[29] (Fig. 3c). Ag nanoparticles can significantly enhance the Raman spectrum of their capping ligands, which is known as the surface-enhanced Raman scattering (SERS) phenomena originating from the dramatic electromagnetic field enhancement around the Ag nanoparticles[30–32]. To investigate the existing status of the dodecyl sulfate ions on the Ag nanoparticles, we performed SERS measurements[33–35]. SERS spectra of the Ag nanoparticles confirmed that the dodecyl sulfate ions were ionically bonded to the surface of the Ag nanoparticles, reflected by the emergence of the 961 cm$^{-1}$ Raman peak associated with silver dodecyl sulfate[21,22] (Fig. 3d). Dodecyl sulfate ions have a hydrophilic sulfate head and a hydrophobic alkyl tail. We supposed that the dodecyl sulfate ions anchored on the Ag nanoparticle surface via non-directional ionic bonds might be able to sense the surrounding liquid medium and rotate to adapt to it. The hydrophobic alkyl tail lies on the Ag nanoparticle surface (referred as lying state) and the hydrophilic sulfate head exposes to water to form electrostatically stabilized colloidal solutions in water. Oppositely, the hydrophobic alkyl tail stretches into oil (referred as standing state) to camouflage the hydrophilic sulfate ions to form sterically stabilized colloidal solutions (Figs. 1c and 3e).

If the orientation of the dodecyl sulfate ions indeed changes, the intensity of the SERS peak of the alkyl tail should vary, because the electromagnetic field (known as hot spot) around the Ag nanoparticles decreases dramatically away from the nanoparticle surface[35] (Fig. 3f). The intensity of the 961 cm$^{-1}$ Raman peak arising from the silver dodecyl sulfate can be used as a standard, because its intensity is not influenced by the orientation change of the dodecyl sulfate ions[33–35]. Experimentally, the intensity ratio between the 1298 cm$^{-1}$ SERS peak originating from the alkyl chain and the 961 cm$^{-1}$ SERS peak of silver dodecyl sulfate was about 1 when the Ag nanoparticles were dispersed in water, while decreasing to be around 0.15 when the Ag nanoparticles were dispersed in oil (Fig. 3g). The prominent intensity decrease of the SERS peak of the alkyl chains when the Ag nanoparticles were transferred from water to oil consolidated that the alkyl chain rotated from the lying state to the standing state[33–35].

We further performed molecular dynamics (MD) simulations to theoretically confirm that the rotation of the ligands can indeed take place[36]. We monitored the variation of the vertical distance (*d*) between the end carbon atom of the dodecyl chain and the Ag surface when exposed to different solvents (Fig. 3h and Supplementary Fig. 17). The ligand at standing state dynamically oscillated at a high frequency of ~0.3 times/ns within the first 100 ns, and then abruptly attached to the Ag surface when exposed to water (red curve in Fig. 3i and Supplementary

Movie 3). The ligand at lying state kept attaching to the Ag surface in water (black curve in Fig. 3i). Experimentally, the Ag nanoparticles in ethanol with their ligands at standing state were indeed immediately dispersible into water (Supplementary Fig. 18). The ligand at lying state stayed attached on the Ag surface when exposed to toluene at 300 K (black curve in Fig. 3j), however, it suddenly evolved to the standing state after 150 ns at 350 K (red curve in Fig. 3j and Supplementary Movie 4). The ligands at standing state dynamically vibrated in toluene at a frequency ~0.24 times/nm (red curve in Fig. 3j and Supplementary Movie 4). The dynamic oscillation of the ligands increased the inter-nanoparticle repulsive force, promoting the colloidal stability. Experimentally, the ligand orientation change from lying to standing status in toluene could take place at room temperature (Supplementary Fig. 18). The Ag nanoparticles in toluene failed to redisperse into water (Supplementary Fig. 18), resulting from the toluene molecule layer on the Ag surface prohibiting the ligand rotation predicted by the MD simulations.

The smart Ag nanoparticles dispersed in water could slowly (~1 month) diffuse into oil, because the alkyl chains on the nanoparticle at the water/oil interface gradually rotated into oil and eventually the whole nanoparticle migrated into oil (Supplementary Fig. 19)[37]. The slow diffusion process from water to oil further proved that the nanoparticles prone to disperse in both water and oil. The slow diffusion process from water to oil might be the origin of the temperature discrepancy between the MD simulations (350 K) and the experimental results (room temperature) of the transition from the lying state to the standing state.

In contrast, Ag nanoparticles dispersed in water prepared by the wet chemical method failed to migrate into the oil phase, instead of forming precipitates at the bottom of water (Supplementary Fig. 19). Injecting ethanol into the water colloidal solution immediately induced aggregation of the nanoparticles at the water/oil interface. The aggregated nanoparticles were transported into the oil phase by shaking the mixture (Supplementary Fig. 20).

The tendency of the shorter alkyl chains rotating to oil phase is weakened and in turn the torque force powering the ligand rotation is insufficient[38]. Therefore, there should be a critical length below which the alkyl chain at the lying state fails to rotate to the standing state to adapt to the oil phase. The experimental results revealed that the alkyl chain should contain at least 10 carbon atoms to realize the ligand rotation (Supplementary Fig. 21). The bonding strength between silver and the sulfate head should be important to realize the orientation change of the dodecyl sulfate ligands. We tried to replace the sulfate head within SDS by other anions, however, precipitates were instantly formed once silver ions met sodium laurate, potassium dodecyl phosphate, and sodium dodecyl sulfonate.

**Antibacterial properties of smart Ag nanoparticles**. Ag nanoparticles have been explored as antimicrobial agents[39–41]. Previous studies revealed that bacteria could develop resistance to Ag nanoparticles by producing sticky proteins or other species, triggering the aggregation of Ag nanoparticles and alleviating or even eliminating their antibacterial activity[41]. Notably, there are still no methods to prohibit flagellum protein (flagellin)-induced aggregation of Ag nanoparticles[41]. Our electrodeposited smart Ag nanoparticles with rotatable surface ligands could disperse in the liquid medium containing flagellin (Supplementary Fig. 22) at a concentration of even up to 2.5 μg/ml for 7 days (Fig. 4a). In contrast, the wet chemistry prepared Ag nanoparticles immediately flocculated once met flagellin (Fig. 4a). Moreover, dodecyl sulfate ligands with similar structure to the bacterial membrane can assist the Ag nanoparticles to enter the bacteria[42,43] (Figs. 1d and 4b).

As anticipated, the electrodeposited smart Ag nanoparticles showed better antimicrobial capabilities than the wet chemistry prepared ones with much lower (usually reduced by more than 10 times) minimal inhibitory concentration (MIC) and minimal bactericidal concentration (MBC) values for the tested 15 bacteria (six gram-positive and nine gram-negative) (Fig. 4c). Meanwhile, the bactericidal process was substantially accelerated (Fig. 4d).

Notably, the dodecyl sulfate ligands without binding to the Ag nanoparticles have very weak antibacterial activity (Supplementary Fig. 23) and they could not enhance the antibacterial performance of silver ions (Supplementary Fig. 24). In contrast, the antibacterial activity of the Ag nanoparticles was dramatically enhanced by the dodecyl sulfate surface ligands (Fig. 4e). The smart Ag nanoparticles showed statistically higher bactericidal capability towards gram-negative bacteria (with thin bacterial membrane and thus easy to pass through by the smart Ag nanoparticles) than towards gram-positive bacteria (with thick bacterial membrane to prohibit the smart Ag nanoparticles from entering). In contrast, Ag nanoparticles prepared by wet chemical methods demonstrated similar antibacterial performance towards the gram-negative and the gram-positive bacteria (Supplementary Fig. 25). These results further proved that the high antibacterial activity of smart Ag nanoparticles arose from the enhanced colloidal stability and their camouflage surface ligands assisting them to enter the bacteria (Fig. 1d).

In practical applications, to wet the surface and deliver the Ag nanoparticles into the small crevices and pores, oil colloidal solutions composed of well-dispersed Ag nanoparticles are preferred. Chemical modifications of Ag nanoparticles, usually associated with valuable and environmentally harmful organic chemicals and solvents, have to be performed to obtain the oil colloidal dispersions. The smart Ag nanoparticles fabricated by a green electrochemical process naturally dispersible in toluene immediately wet the wood surface and delivered the Ag nanoparticles into the sub-microscale channels of the wood verified by the dense distribution of Ag nanoparticles on the longitudinal section of the wood (Fig. 4f, Supplementary Fig. 26, and Supplementary Movie 5). The smart Ag nanoparticles dispersed in water could not completely wet the wood surface (Fig. 4g). Even worse, aqueous colloidal solutions could not even attach to the surface of the vertically placed wood (Supplementary Fig. 27 and Supplementary Movie 6). After rinsing with water or grinding with a sand paper, the wood contaminated by the oil colloidal solutions composed of well-dispersed smart Ag nanoparticles still showed high antimicrobial activity whether in cultivation medium (Fig. 4h) or in air (Supplementary Fig. 28).

## Discussion

In this work, we have demonstrated an approach to break the nanoparticle's dispersible limit via rotatable surface ligands,

allowing the smart nanoparticles to disperse in a broad range of liquids. The smart Ag nanoparticles can be continuously electrodeposited in the electrolyte solution, which turns the electrode-confined low throughput electrodeposition method into a high efficiency and massive smart nanoparticle fabrication approach. The constructed automatic electrodeposition apparatus further puts forward the practical application process of our smart nanoparticles. The good colloidal stability and bacterial membrane-mimicking surface ligands of the smart Ag nanoparticles endow them with high antimicrobial activity. Future efforts will devote to shape and size control of the smart Ag nanoparticles. We envision that rotatable surface ligands can be extended into nanoparticles of other materials, bearing promising applications in colloidal chemistry, optoelectronic devices, 3D printing, and biomedical fields.

## Methods

**Preparation of the Ag nanoparticles**. The smart Ag nanoparticles were electrodeposited in a two-electrode apparatus. The anode was a carbon rod with a diameter of ~5 mm. The cathode was an aluminum foil with a dimension of 25 mm × 75 mm. The deposition voltage was 30 V and the deposition time was 5 min, unless otherwise specified. The electrolyte was composed of 30 mM silver nitrate and 90 mM SDS, unless otherwise specified. The electrodeposition was performed at room temperature (~25 °C).

The wet chemical method used to prepare Ag nanoparticles was carried out in an ice water bath to slow down the reaction process. The sodium borohydride was used to reduce silver nitrate. The concentration of the silver nitrate and the sodium borohydride was 5 and 20 mM, respectively. 50 ml of silver nitrate aqueous solutions was added drop by drop to the sodium borohydride solution during stirring within 2.5 h. The slow dropping process is critical and otherwise the solution would instantly turn dark induced by the aggregation of the formed Ag nanoparticles. After the titration process, the solution was stirred until it cooled to room temperature. The solution was then heated to 80 °C, and naturally cooled to room temperature. The wet chemical or electrodeposited Ag nanoparticles were separated from the solution by centrifugation at $6037 \times g$ for 15 min.

**Preparation of the Au nanoparticles**. Au nanoparticles with a diameter of 45 nm were synthesized by a seed-mediated growth method. Briefly, 0.6 ml of fresh $NaBH_4$ solutions at 10 mM concentration were rapidly added into 5 ml solutions containing of $HAuCl_4$ (0.25 mM) and cetyltrimethyl ammonium bromide (CTAB) (100 mM). After stirring at $100 \times g$ for 2 min, the CTAB-capped Au clusters were obtained. The solution was aged at 27 °C for 3 h. Then, cetyltrimethyl ammonium chloride (CTAC)-capped Au seeds with a diameter of 9 nm were synthesized. 4 ml of aqueous solutions of CTAC (200 mM), 3 ml of ascorbic acid (AA) (100 mM), and 80 μl of CTAB-capped Au cluster dispersions were mixed thoroughly in a glass vial, followed by one-shot injection of 4 ml of aqueous $HAuCl_4$ solutions (0.5 mM). The reaction continued at 27 °C for 15 min. The products were collected by centrifugation at $6708 \times g$ for 15 min. Uniform Au nanoparticles with a diameter of 45 nm were obtained.

**Characterization**. The morphology of the Ag nanoparticles was observed with a transmission electron microscopy (TEM) (HT770) operating at 80 kV and a scanning electron microscope (SEM) (Hitachi SU-8010) operating at 5.0 kV. The microstructure of the electrodeposited Ag nanoparticles was determined by an X-ray diffraction spectrometer (XRD) (D/Max 2550 pc). The formation efficiency of the Ag nanoparticles was estimated using a thermal gravimetric analyzer (DSCQ1000) through annealing the colloidal solution under the protection of nitrogen. All of the photographs were recorded by a digital camera (Canon EOS750D). SERS measurements were performed on a confocal Raman microscopic system (LabRAM HR Evolution, France). The excitation wavelength was 633 nm generated by an Nd: yttrium-aluminum-garnet laser operating at a power of about 0.25 mW. The accumulation time for the SERS signals was 10 s. Fourier transform infrared spectroscopy (FTIR) and X-ray photoelectron spectroscopy (XPS) measurements were operated on a Nicolet 5700 FTIR spectrometer and an Axis Supra XPS spectrometer, respectively. The absorption spectra were obtained using an ultraviolet-visible-near infrared spectrophotometer (PerkinElmer Lambda 950).

**Estimation of the formation efficiency of the Ag nanoparticles**. The thermogravimetric (TG) analysis was used to estimate the particle formation efficiency of the Ag nanoparticles (Supplementary Fig. 4). After heating colloidal solutions (the mass was marked in Supplementary Fig. 4) at 600 °C under nitrogen protection, only Ag nanoparticles and the decomposed product of SDS left. We estimated the weight of the decomposed product of SDS by heating SDS powder. Therefore, the weight of Ag nanoparticles within the colloidal solution was obtained. Then, the production efficiency was estimated by the weight of Ag nanoparticles within the colloidal solution over the preparation time.

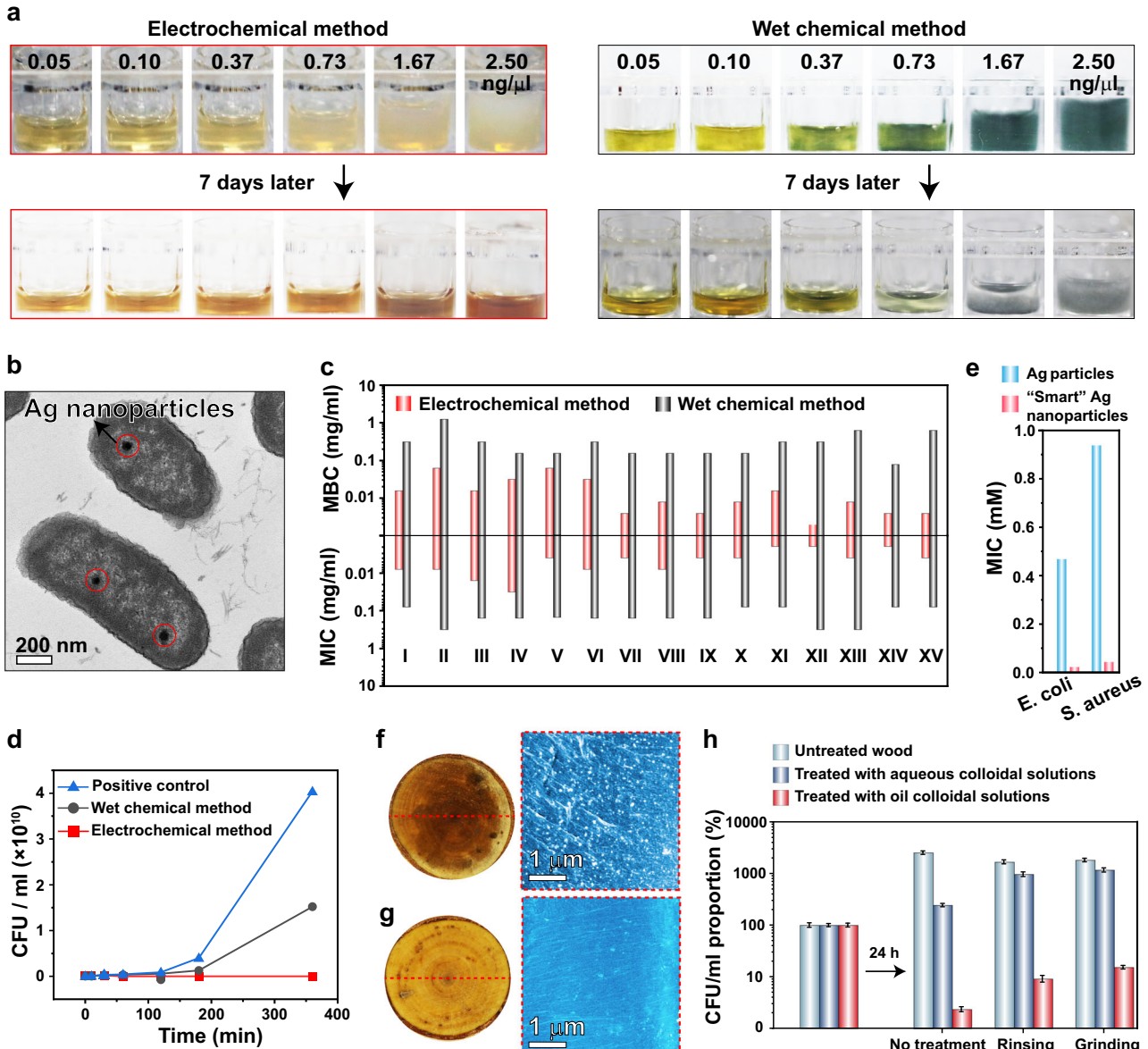

**Fig. 4 Antibacterial performance of the smart Ag nanoparticles. a** Introducing different amounts of flagellin into the aqueous colloidal solutions composed of smart Ag nanoparticles and wet chemistry prepared ones. **b** Smart Ag nanoparticles were observed within the *E. coli* during the bactericidal process. **c** MIC and MBC of the smart Ag nanoparticles and wet chemical method prepared ones toward different bacteria. I: *S. aureus* MY0184, II: *E. faecium* SAL05040, III: *E. faecium* SAL05041. IV: *S. pneumoniae* MY0312. V: *L. monocytogenes* L0019. VI: *L. monocytogenes* L0020. VII: *S. Newport* MY0180. VIII: *S. Typhimurium* MY0198. IX: *S. Enteritidis* MY0199. X: *P. aeruginosa* MY1082. XI: *E. coli* MY1083. XII: *K. pneumoniae* SAL05042. XIII: *K. pneumoniae* SAL05043. XIV: *Y. enterocolitica* MY0125. XV: *Y. pseudotuberculosis* MY0126. The first six are gram-positive and the others are gram-negative bacteria. **d** Sterilization kinetic curves of smart Ag nanoparticles and wet chemistry prepared ones. CFU/ml means colony-forming units per milliliter. **e** Antibacterial effect of the electrodeposited Ag particles scratched from the electrode surface without dodecyl sulfate surface ligands and smart Ag nanoparticles. **f**, **g** Photos (left) of the top and scanning electron microscopy images (right) of the longitudinal section (5 mm below the top surface) of the wood treated by smart Ag nanoparticles (white dots) dispersed in toluene and water, respectively. **h** Antimicrobial performance of the wood stained by aqueous and toluene colloidal solutions composed of smart Ag nanoparticles without treatment, after water rinsing, and sand paper grinding treatment. Error bars represent standard deviation on the basis of five independent measurements.

**Continuous electrodeposition of smart Ag nanoparticles**. The smart Ag nanoparticles were electrodeposited in the electrolyte solution. Then, the solution was transported to the filtration setup using a water pump (Kamoer DIPump550) operating at a power of ~50 W. The smart Ag nanoparticles were trapped by the ultrafiltration membrane with a pore size of 10 nm when the electrolyte solution was forced to pass through the ultrafiltration membrane using a mechanical pump (WIGGENS V300). The filtered electrolyte was transported back to the electrodeposition cell using another water pump.

**Electrochemical pen**. The electrochemical pen was composed of a syringe with a volume of 10 ml. A copper wire was inserted to the electrolyte filled in the syringe

behaving as the anode electrode. The other copper wire was fixed about 1 mm away to the needle behaving as the cathode electrode. Squeezing the syringe would form an electrolyte droplet hanging on the tip of the needle to contact the cathode copper wire, generating Ag nanoparticles immediately under applied potentials. Slowly moving the needle on any arbitrary substrate would create conductive paths.

**Molecular dynamics simulations**. All-atom molecular dynamics (MD) simulations were carried out for a single dodecyl sulfate ion (ligand) attached on Ag flat surface, using GROMACS 2020.6 package[44]. The equations of motion were integrated with the leapfrog algorithm using a time step of 1.0 fs. The temperature was controlled by the v-rescale method with a damping constant of 1.0 ps. The

rhomboic simulation cell was used ($a = 3.2513$ nm, $b = 3.2513$ nm, $c = 1.9827$ nm, $\alpha = \beta = 90°$, and $\gamma = 120°$). A periodic boundary condition was applied to all the directions. The Ag plate (432 atoms), extending on the $xy$ (=$ab$) plane, was composed of three (111) layers of the face-centered cubic lattice, whose parameters were taken from AMCSD database[45]. An oxygen atom of the ligand was bound to one of the Ag atoms, with a distance of 0.36 nm. The solvent was either of water (934 molecules) or toluene (187 molecules). Ag atoms were reproduced by the force field proposed by Hendrik et al.[46], while water molecules were reproduced by the TIP4P/2005 model[47]. Toluene and ligand molecules were represented by the OPLS-AA force fields[48] except the charges determined by the CM1-LBCC charge model[49], and both of which were obtained from the LigParGen website[50].

In each solvent, two different initial conformations of the ligand were considered, i.e., standing and lying. The solvent molecules were randomly poured into the ligand-side of the simulation box, while the remaining was vacant. The initial conformation was first energy-minimized using the steepest descent algorithm. Then, at the temperature of 300 and 350 K, an equilibration MD simulation for 100 ps was followed by a 500 ns production run, both in the canonical (NVT) ensemble. In order to confirm the results obtained for the initially lying ligand in toluene at 300 and 350 K, we sampled three independent trajectories starting from different conformations.

**Bacterial strains**. Bacterial strains *S. aureus* MY0184, *S. pneumoniae* MY0312, *L. monocytogenes* L0019, *L. monocytogenes* L0020, *S. Newport* MY0181, *S. Typhimurium* MY0198, *S. Enteritidis* MY0199, *P. aeruginosa* MY0182, *E. coli* MY0183, *Y. enterocolitica* MY0125, and *Y. pseudotuberculosis* MY0126 were purchased from Sino Zhongyuan, China. Bacterial strains *E. faecium* SAL05040, *E. faecium* SAL05041, *K. pneumoniae* SAL05042, and *K. pneumoniae* SAL05043 were exacted from the anal swabs of sick pets in hospital of Zhejiang University.

**Antibacterial paints**. The trunk of birch tree with a diameter of 50 mm was cut into 5-mm-thick cylinder pieces. 5 ml of aqueous and oil colloidal solutions composed of 25 mg smart Ag nanoparticles were sprayed onto the wood surface. Then, the wood pieces with and without smart Ag nanoparticles were contaminated by bacteria. Briefly, 100 μl of *E. coli* suspensions at a concentration of $10^8$ CFU/ml were evenly seeded on the surface of the wood pieces with and without smart Ag nanoparticles. Then, the wood pieces were cultured at 37 °C for 24 h in the culture medium. Swabbing the wood surface for 3 times with a sterile absorbent cotton swabs and resuspending the cotton swabs in sterile phosphate buffer saline (PBS) environment to evaluate the number of the bacteria on different wood pieces. After serial dilution of the above PBS buffer solutions, they were introduced into the Brain Heart Infusion (BHI) plate. The standard plate count method was used to calculate the bacterial colonies per milliliter. Through monitoring the number variation of the bacteria on the wood in dry and the culture medium, the antibacterial performance of the aqueous and the oil colloidal solution paints was evaluated. To further evaluate the robustness of the antibacterial paints, the wood pieces stained by aqueous and oil colloidal solutions were rinsed with 10 liter of water or grinded with a sand paper for 100 times. Then, their antibacterial activity was evaluated.

**Measurements of MIC and MBC**. MIC and MBC of the electrodeposited smart and wet chemistry prepared Ag nanoparticles were determined according to the National Committee for Clinical Laboratory Standards (NCCLS) of antimicrobial susceptibility test. Mueller-Hinton broth (pH value: 7.2–7.4) was used as the culture medium. The fastidious bacteria *L. monocytogenes* was cultured in BHI broth. Three to five colonies with similar morphology were selected from the inoculating ring and inoculated in 5 ml of Mueller-Hinton broth at 37 °C for 2 to 6 h. The logarithmic growth period after enrichment was corrected by Mueller-Hinton broth to 0.5 McFarland standard, containing about $2 \times 10^8$ CFU/ml bacteria. The above bacteria suspension was diluted by 100 times by Mueller-Hinton broth for standby. The bacterial suspension with different concentrations of electrochemically and wet chemically prepared Ag nanoparticles were inoculated in 96 well plates, and cultured at 37 °C for 24 h. The MIC is the lowest concentration of Ag nanoparticles to make the bacterial solution clear. Bacterial suspensions including one, two, four, and eight times of MIC were added to Mueller-Hinton agar. The *L. monocytogenes* was added to the BHI agar. The MBC is the lowest concentration of Ag nanoparticles to prevent the bacteria growth on the plate. The experiments were performed three times to obtain reliable MIC and MBC.

**Sterilization kinetic curves of Ag nanoparticles**. *L. monocytogenes* L0019 was selected as the test bacteria because of the smallest difference of MBC between the smart Ag nanoparticles and wet chemistry prepared ones. The bacterial suspensions were treated with electrochemically and wet chemistry prepared Ag nanoparticles. The concentration of Ag nanoparticles in each solution was ~0.15625 mg/ml, which is the MBC value of wet chemistry prepared Ag nanoparticles towards *L. monocytogenes* L0019. The initial concentration of the bacterial solution was adjusted to be ~$6 \times 10^7$ CFU/ml. The number of the bacteria was counted according to the standard plate count method after diluting the bacterial suspensions with sterile PBS buffer. Each experiment was performed three times to guarantee the reliability.

**Preparation of bacteria samples for TEM characterization**. Two milliliters of *E. coli* suspensions ($2 \times 10^8$ CFU/ml) were cultured with 100 μl of Ag nanoparticle colloids for 24 h. After centrifuging the suspensions at $2012 \times g$ for 10 min and removing the supernatant, the precipitates were dissolved in 2.5% glutaraldehyde solutions and fixed at 4 °C for 12 h. The fixed samples were consequently dehydrated for 15 min with ethanol at gradually increased concentrations (e.g., 30%, 50%, 70%, and 80%). Then, they were treated for 15 min sequentially by 90% and 95% acetone solutions, and by pure acetone twice for 20 min. Further, the samples were consecutively treated with a mixture of Spurr embedding agent and acetone at a volume ratio of 1:1 for 1 h and a mixture of the two at a volume ratio of 3:1 for 3 h. The samples were stored overnight in pure embedding agent at room temperature. The samples embedded with bacteria were cut into 70 to 90 nm thin slices using the Leica EM UC7 ultrathin slicer. The slices were stained with lead citrate solution and 50% saturated ethanol solution of uranium dioxide acetate for 5 to 10 min, respectively. The thin slices were observed by a Hitachi H-7650 TEM microscope.

**Preparation of the flagellin**. One hundred microliters of the adjusted bacterial strain [*Salmonella* enterica serovar Typhimurium (ATCC14028) wild type] was pre-cultured in 10 ml of Luria-Bertani (LB) medium (1:100 dilution) with OD600 (absorption value at 600 nm) equals 1. The samples were incubated until the OD600 value approximately reached 0.6–0.8. The following steps were performed on ice. Five milliliters of each normalized sample was placed in an individual sterile petri dish plate and then the inoculum was lifted up from the dish plate with a sterile syringe. The cell suspensions were injected back into the petri dish plate with a sterile syringe. The previous steps were repeated for 30 times to shear off as much flagella as possible. One milliliter of the cell suspensions was transferred to an Eppendorf tube and was centrifuged at $6708 \times g$ for 10 min at 4 °C. Eight hundred microliters of the supernatant was filled to a new centrifugation tube and centrifuged at $6708 \times g$ for 10 min at 4 °C. Seven hundred and fifty microliters of the centrifugal separations were transferred to another centrifugation tube and gently mixed with cold trichloroacetic acid solutions. Then, the samples were placed on ice or stored in a −20 °C freezer for 12 h. Eventually, the flagellin was obtained by centrifugation at $6708 \times g$ for 40 min at 4 °C. The molecular weight of the flagellin determined by the protein gel electrophoresis verified the successful preparation of flagellin.

**Estimation of the aggregation resistance of Ag nanoparticles to flagellin**. We added 500 μl of the Ag colloidal solutions composed of smart Ag nanoparticles and wet chemistry prepared ones to two lines of orifice plates, respectively. The flagellin solutions at varied concentrations were added to the colloidal solutions composed of smart Ag nanoparticles and wet chemistry prepared ones, respectively. The mixtures were placed at room temperature for 7 days to observe the stability.

**Reporting summary**. Further information on research design is available in the Nature Research Reporting Summary linked to this article.

## Data availability
All experimental data within the article and its Supplementary Information are available from the corresponding authors upon request.

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

## Acknowledgements

We acknowledge generous funding support from the Zhejiang Provincial Natural Science Foundation of China (LR19E010001 and LR19C180001), National Key Research and Development Program of China (2018YFB0703803 and 2019YFE0103900), and National Natural Science Foundation of China (51971200).

## Author contributions

S.Y. conceived the idea. S.Y., M.Y., K.M., and Z.W. designed the experiments. Y.L., L.Z., J.W., and L.P. carried out the material synthesis, structural, and other characterizations. N.P. and X.P. performed the antibacterial experiments. Y.Y., X.Z., and K.M. performed the molecular dynamics simulations. All authors analyzed the data and contributed to writing of the manuscript.

## Competing interests

The authors declare no competing interests.
