## [Peer Review File · Nature Communications]

REVIEWER COMMENTS

Reviewer #1 (Remarks to the Author):

The manuscript introduces an interesting concept of nanoparticles with surface-rotatable ligands, which in turn can induce surface adaptation to enhance cell penetration and antibacterial activity, shown for the example of SDS-modified silver nanoparticles in various solvents.

The production of Ag nanoparticles in the way described by the authors seems effective and could be potentially employed by others. However, how good is the control over nanoparticle size and surface structure in this method? Are there significant differences as a function of voltage? How critical is the use of DDS, and could other surfactants, e.g. with shorter or longer alkyl chains also be used? How much would the results differ? The average particle size is given, however, quantitative data on surface topography appear to be missing and would be nice to validate and know more quantitatively.

The slow dissolution/dispersion of the particle into oil phases is well described, and similar phenomena of surface relaxation of alkyl ligands are known, for example, on clay surfaces (<https://pubs.acs.org/doi/abs/10.1021/ja021248m>). There clearly are different barriers and dynamic time scales at play. The simulations give some qualitative insight into the surfactant orientation. However, I think more solvent would need to be included (Figure 3h-j) unless the particles would be used in vacuum. These mechanistic insights appear to be a bit underdeveloped and can be improved. The models and simulations themselves should be accurate enough to obtain more detailed insights, i.e., by using more realistic model setups with full 3D solvent (and maybe a lipid interface as a membrane model).

Overall, I find the experimental demonstrations interesting although at the same time the storyline and clarity of messages can be much improved. I think the hypothesis and overall purpose of the study gets a bit buried in the details. It is hard to follow some of the arguments and the manuscript would benefit from much stronger and clear connections between the different parts and observations.

I would therefore recommend a major revision to better explain the key features of the proposed mechanisms and relationship to antibacterial function. Maybe a new schematic is needed in Figure 1, or a new Figure 1. I would let the authors decide.

The study remains, in principle, interesting but should be more crystallized and better explained in logical/linear order to become clear and easy to follow for the readers. The possible design space and limitations should also be mentioned in the main text.

Reviewer #2 (Remarks to the Author):

The authors present a study aimed on the versatile dispersion of silver NPs in various polar and nonpolar solvents based on the surface modification using SDS. The results of this study will undoubtedly be beneficial not only in basic research, but especially in applied research, where the long-term aggregate and sedimentation stability of silver particles in various solvents and liquid media plays a crucial role and ultimately determine whether nanosilver and its specific chemical, biological, catalytic, etc. properties will be applicable or not in the given practical application. I propose to accept the work after the following minor revisions.

1) it would be highly desirable and useful for authors to verify the long-term stability (at least 7 days) of silver nanoparticles in all the solvents that they applied, as long-term stability is important for practical applications, especially biological ones.

2) According to the Figure 3a it is clear that the optical properties (absorption spectra) change slightly, especially in non-polar solvents. Depending on the course of the spectrum, I deduce changes in either the morphology of nanosilver or its aggregate instability. It would be desirable to verify by characterization by electron microscopy whether nanosilver does not change its morphology or does not partially aggregate after transfer to a non-polar environment. Please include TEM imaged of silver NPs dispersed in all used solvents.

3) Wet chemical methods are still among the most used methods of nanosilver preparation both for their simplicity and economic and time cost, but mainly for the ability to control the size and especially the shape and morphology of produced silver nanoparticles in contrast to limited controllable electrochemical preparation methods. Is it possible to use this method of nanosilver modification and the ability to disperse in polar and non-polar solvents also in the case of silver nanoparticles prepared by wet chemical reduction methods? If so, then it would greatly expand the possibilities of applying silver nanoparticles in practice. Given the ability to control the wet reduction process and purposefully control the size and shape of nanoparticles, this would be very beneficial and would move the synthesis options forward, as well as application possibilities.

Responses to Referees' comments

Reviewer #1

Overall Comment: The manuscript introduces an interesting concept of nanoparticles with surface-rotatable ligands, which in turn can induce surface adaptation to enhance cell penetration and antibacterial activity, shown for the example of SDS-modified silver nanoparticles in various solvents.

Author Response: Thank you for your positive and constructive comments, which are all valuable and very helpful for revising and improving our paper. We have studied and addressed the comments carefully and revised the manuscript according to your suggestions/comments.

Comment 1: The production of Ag nanoparticles in the way described by the authors seems effective and could be potentially employed by others. However, how good is the control over nanoparticle size and surface structure in this method? Are there significant differences as a function of voltage? How critical is the use of SDS, and could other surfactants, e.g. with shorter or longer alkyl chains also be used? How much would the results differ? The average particle size is given, however, quantitative data on surface topography appear to be missing and would be nice to validate and know more quantitatively.

Response 1: Thank you for your comments. To control the nanoparticle size and surface structure, it is critical to figuring out the formation mechanism of Ag nanoparticles. The Ag nanoparticles were first formed on the electrode surface, and then were detached from the electrode surface and diffused into the solution (Figure R1a). Therefore, the size of the Ag nanoparticles is determined by their growth speed and their staying time on the electrode surface. The growth speed is mainly determined by the voltage, while the staying time on the electrode surface is dependent on the value of the adhering force (f_3) and the detaching force (f_1 and f_2) (Figure R1b). The shear force f_1 is caused by the electrodynamic force and the bubble formation/release from the electrode surface (mainly related to the deposition voltage), and the electrostatic repulsive force f_2 arises from the repulsion between the negatively charged Ag nanoparticles and the cathode surface (mainly related to the SDS concentration). Overall, the deposition voltage and the concentration of SDS will

influence the size and the surface structure of the Ag nanoparticles. Generally, the faster the Ag nanoparticles are detached from the cathode surface, the smaller and more spherical the Ag nanoparticles would be.

Fig. R1. The formation mechanism of Ag nanoparticles. **a**, Schematic illustration of the formation process of Ag nanoparticles. Process I: the nuclei are formed on the electrode surface passivated by dodecyl sulfate ions. Process II: The nanoparticles are negatively charged induced by the dodecyl sulfate ligands. Process III: The Ag nanoparticles are detached from the electrode surface. **b**, Schematic diagram of forces exerted to the Ag nanoparticles on the electrode surface. f_1 , f_2 , and f_3 represent the shear force induced by the electrodynamic force and the formation/release of hydrogen bubbles, the electrostatic repulsive force between the nanoparticle and the electrode, and the adhesive force between the Ag nanoparticle and the electrode, respectively.

Based on the above understanding, we experimentally and theoretically studied the size and surface structure evolution at different deposition voltages and SDS concentrations. As the voltage was increased, both f_1 and f_2 were enhanced, significantly reducing the growth time of the Ag nanoparticles. Therefore, the size of the Ag nanoparticles was reduced (Figure R2). Experimentally, the mean size of the Ag nanoparticles was reduced from 50 nm to 34 nm, 26 nm, and 12 nm when the voltage was increased from 10 V, to 20 V, 30 V, and 100 V, respectively (Figure R2).

Fig. R2. Morphology and size distribution of the “smart” Ag nanoparticles synthesized at different voltages. a to d, TEM images of Ag nanoparticles synthesized at 10 V, 20 V, 30 V, and 100 V, respectively. e to h, Size distribution of the Ag nanoparticles based on the measurement of > 100 nanoparticles synthesized at 10 V, 20 V, 30 V, and 100 V, respectively. Inset: Tyndall effect of the colloidal solutions.

When the SDS concentration was increased, f_2 was greatly enhanced. Therefore, the size of the Ag nanoparticles was reduced from 70 nm, to 41 nm, 30 nm, and 30 nm when the SDS/ Ag^+ molar ratio was increased from 0.5, to 1.5, 3, and 4.5, respectively (Figure R3).

Fig. R3. TEM images and size distribution of the “smart” Ag nanoparticles electrodeposited in the electrolyte with different molar ratios of SDS to silver nitrate. a to d, The molar ratio of SDS to silver nitrate is 0.5, 1.5, 3, and 4.5, respectively. e to h, Size distribution of the Ag nanoparticles based on the measurement of > 100 nanoparticles electrodeposited in the electrolyte with different molar ratios of SDS to silver nitrate (30 mM) at 0.5, 1.5, 3, and 4.5, respectively.

The nanoparticles turned to be more spherical when the SDS concentration and the electrodeposition voltage were increased (Figure R2 and R3). To qualitatively

describe the surface topography of the Ag nanoparticles, we performed HRTEM (High Resolution Transmission Electron Microscopy) characterization of the Ag nanoparticles prepared at different deposition voltages (Figure R4). The shape profile and the surface roughness of the Ag nanoparticles were analyzed. The shape profile of the Ag nanoparticles could be obviously observed from the HRTEM images shown in Figure R4a to c. These images showed that the Ag nanoparticles turned to be more spherical as the voltage was increased.

Considering that the Ag nanoparticles fabricated at different voltages were almost spherical, we chose the circularity as a representative parameter to further quantitatively describe the surface topography. The circularity is determined by the projected area (A) of the Ag nanoparticles and the overall perimeter (L) of the projection according to the formula (see *Journal of Paleontology*, 1927, 1, 179-183):

$$\omega_{\text{circularity}} = 4\pi A / L \quad (1)$$

Surface roughness of Ag nanoparticles is reflected in this shape factor due to the incorporation of the overall perimeter. $\omega_{\text{circularity}}$ is dependent on the shape of the projected image (Figure R4d to f), being 1 for spherical spheres and less than 1 for any other shapes (Figure R4g). When the voltage was increased, the $\omega_{\text{circularity}}$ value was also increased, indicating the reduced surface roughness. These results have been included in the revised manuscript.

Fig. R4. Surface topography of Ag nanoparticles prepared at different voltages. **a to c**, HRTEM images of the Ag nanoparticles dispersed in aqueous solutions prepared at 10 V, 20 V, and 30 V, respectively. **d to e**, The representative projections of the HRTEM images of Ag nanoparticles prepared at 10 V, 20 V, and 30 V, respectively. **g**, Quantitatively characterization of the surface roughness of the Ag nanoparticles prepared at different voltages.

Without SDS in the electrolyte solution during electrodeposition, no Ag nanoparticles were generated in the electrolyte solution. Instead, a black film was formed on the electrode surface (see the inset image in Figure R5a).

To verify whether the type of the surfactant would affect the preparation and dispersible properties of the electrodeposited Ag nanoparticles, we replaced the anionic surfactant SDS with neutral surfactant PVP. Ag nanoparticles could also be continuously and massively fabricated (Figure R5a and b). However, the dispersible property of Ag nanoparticles prepared using PVP was different from that of the Ag nanoparticles prepared using SDS, which could not form stable colloidal solutions whether in water or in organic liquids (Figure R5c and d).

Fig. R5. The influence of surfactants on the formation of Ag nanoparticles. a, Electrochemical fabrication of Ag nanoparticles in the electrolyte composed of PVP. Yellow mist was formed during electrodeposition at 30 V. Inset: Electrodeposition of Ag particles in 30 mM silver nitrate electrolyte solution without any surfactants. **b,** Photo of the colloidal solution and the corresponding TEM image of the fabricated Ag nanoparticles. Inset: Magnified image. **c,** Ag nanoparticles precipitated at the bottom after 7 days. **d,** Aggregates immediately formed when freshly prepared Ag nanoparticles were transferred into toluene. **e,** The influence of the alkyl chain length on the dispersible property of the Ag nanoparticles. The alkyl chain should contain at least 10 carbon atoms to enable the Ag nanoparticles to disperse in both water and oil. Otherwise, the Ag nanoparticles cannot disperse in neither water nor

The alkyl chain length of the surfactant molecules also has a great impact on the dispersibility of the Ag nanoparticles. Experimentally, we studied the alkyl chain composed of 16, 12, 10, 8, and 6 carbon atoms (Figure R5e). When the alkyl chain was short, its tendency to the oil phase was weak. Therefore, the ligand rotation could not take place. We found that the alkyl chain should at least contain 10 carbon atoms to realize the ligand rotation and enable the Ag nanoparticles to disperse in both water and organic liquids (Figure R5e).

We also replaced the sulfate head of SDS by other anions, precipitates were instantly formed once silver ions met sodium laurate, potassium dodecyl phosphate, and sodium dodecyl sulfonate, failing to obtain stable electrolyte solutions.

Your comments have greatly improved the quality of our work. We have added the above contents into our revised manuscript.

Comment 2: The slow dissolution/dispersion of the particle into oil phases is well described, and similar phenomena of surface relaxation of alkyl ligands are known, for example, on clay surfaces (<https://pubs.acs.org/doi/abs/10.1021/ja021248m>). There clearly are different barriers and dynamic time scales at play. The simulations give some qualitative insight into the surfactant orientation. However, I think more solvent would need to be included (Figure 3h-j) unless the particles would be used in vacuum. These mechanistic insights appear to be a bit underdeveloped and can be improved. The models and simulations themselves should be accurate enough to obtain more detailed insights, i.e., by using more realistic model setups with full 3D solvent (and maybe a lipid interface as a membrane model).

Response 2: We regret our descriptions were misleading. In our MD simulations, the ligand was fully dissolved in solvent but we only showed a part of solvent molecules in Figure 3h-j in order to highlight the orientation change of the ligand. We added the following sentence in the caption of Figure 3 with a new figure (Fig. R6) in the revised Supporting Information.

“In the insets, only the solvent molecules on the Ag surfaces were shown, while the ligand was fully immersed in the solvent in MD simulations.”

Fig. R6. Snapshots of the whole systems used in MD simulations. a, b, The ligand at standing state on Ag surface in toluene and water, respectively. **c, d,** The ligand at lying state on Ag surface in toluene and water, respectively.

The reference you mentioned is really helpful and we have cited the paper in the revised manuscript. The slow dispersion of the Ag nanoparticles into the oil phase is because of the slow orientation of the ligand to the oil phase. During the simulation, we realized the orientation change of the ligand at 350 K. We have added these contents into the revised manuscript.

Comment 3: Overall, I find the experimental demonstrations interesting although at the same time the storyline and clarity of messages can be much improved. I think the hypothesis and overall purpose of the study gets a bit buried in the details. It is hard to follow some of the arguments and the manuscript would benefit from much stronger and clear connections between the different parts and observations. I would therefore recommend a major revision to better explain the key features of the proposed mechanisms and relationship to antibacterial function. Maybe a new schematic is needed in Figure 1, or a new Figure 1. I would let the authors decide. The study remains, in principle, interesting but should be more crystallized and better explained in logical/linear order to become clear and easy to follow for the readers. The possible design space and limitations should also be mentioned in the main text.

Response 3: Thank you for your valuable suggestions. We have refined the storyline and removed some of the experimental details into the supporting information part. To clearly show the hypothesis and the overall purpose of the study, we updated the Figure 1 (the schematic) in the revised manuscript covering the ligand rotation, dispersibility, and the antibacterial mechanism (Figure R7). We added sub-section titles to strengthen the connection between different parts and observations in the revised manuscript per your suggestions. The revised manuscript has been more crystallized and better explained in logical/linear order according to your valuable suggestions. The revised manuscript should be easy to follow for the readers. The limitations (mainly shape and size control of the nanoparticles) and potential of the design method have also been discussed in the conclusion part of the revised manuscript.

Fig. R7. Existence of a dispersible limit for specific nanoparticles. **a**, Different strategies used to stabilize nanoparticles in different solvents. (I) Electrostatically stabilized nanoparticles in polar solvents. (II) Sterically stabilized nanoparticles in nonpolar solvents. (III) Amphiphilic ligands stabilized nanoparticles. (IV) Mixed ligands stabilized nanoparticles. **b**, The appropriate dispersible limit of different stabilizing strategies shown in **a** and (V) breaking the dispersible limit of common stabilizing strategies by tethering rotatable surface ligands onto nanoparticles. The rotatable surface ligands allow the nanoparticles to be electrostatically stabilized in polar solvents, while sterically stabilized in nonpolar solvents. **c**, Reversible orientation change of the rotatable ligands to adapt to surrounding liquids. **d**, The antibacterial performance of the “smart” Ag nanoparticles is enhanced by the fact that they do not aggregate and they can easily penetrate the bacterial membrane.

Reviewer #2

Overall Comment: The authors present a study aimed on the versatile dispersion of silver NPs in various polar and nonpolar solvents based on the surface modification using SDS. The results of this study will undoubtedly be beneficial not only in basic research, but especially in applied research, where the long-term aggregate and sedimentation stability of silver particles in various solvents and liquid media plays a crucial role and ultimately determine whether nanosilver and its specific chemical, biological, catalytic, etc. properties will be applicable or not in the given practical application. I propose to accept the work after the following minor revisions.

Author Response: We are very grateful for your positive comments and valuable suggestions, which have greatly improved the quality our work. We have addressed your comments and revised the manuscript accordingly.

Comment 1: It would be highly desirable and useful for authors to verify the long-term stability (at least 7 days) of silver nanoparticles in all the solvents that they applied, as long-term stability is important for practical applications, especially biological ones.

Response 1: Thank you for your constructive comments. Based on your suggestion, we studied the stability of the silver nanoparticles in different solvents for 7 days (Figure R1). After 7 days of storage, no precipitations were observed in all of the silver colloidal solutions (Figure R1), indicating the outstanding colloidal stability. The SEM images proved that the Ag nanoparticles were still separated from each other after 7 days. We will explore the applications of these stable colloids in different fields in the future.

Fig. R1. Long-term stability of the silver nanoparticles dispersed in different liquids. **a**, Photos of silver colloidal solutions stored at room temperature for at least 7 days. **b**, Tyndall effect and TEM images of the silver nanoparticles dispersed in different solvents for 7 days.

Comment 2: According to the Figure 3a it is clear that the optical properties (absorption spectra) change slightly, especially in non-polar solvents. Depending on the course of the spectrum, I deduce changes in either the morphology of nanosilver or its aggregate instability. It would be desirable to verify by characterization by electron microscopy whether nanosilver does not change its morphology or does not partially aggregate after transfer to a non-polar environment. Please include TEM imaged of silver NPs dispersed in all used solvents.

Response 2: Thank you for your comments. The absorption peak of the silver nanoparticle colloids arises from the localized surface plasmon resonance (LSPR), induced by the collective oscillation of free electrons under light irradiation. The location of LSPR is related the size, shape, and the dielectric constant of the surrounding environment. According to your comments, the TEM images of silver nanoparticles dispersed in different solvents were exhibited (Figure R1b). No obvious changes in morphology or size of the silver nanoparticles, or the aggregation were

observed. Since we separated the Ag nanoparticles from the solvent by the centrifugation method, slight size change was expected. However, the obvious LSPR peak shift in different solvents should be mainly induced by the change of the refractive index. The LSPR peak shift ($\Delta\lambda$) in response to changes in the refractive index (n) is approximately described by (*Nature Materials*, 2008, 7, 442-453):

$$\Delta\lambda \approx m(n_1-n_2)(1-e^{-2d/l}) \quad (1)$$

Where m is the sensitivity factor (in nm per refractive index unit), n_1 and n_2 are the refractive indices (in nm per refractive index unit) of the adsorbate and medium surrounding the nanoparticles, respectively, d is the effective thickness of the adsorbate layer (in nm), and l is the electromagnetic field decay length (in nm). The LSPR peak of silver nanoparticles dispersed in non-polar solvents could be affected by many factors, especially the adsorbate and medium surrounding the nanoparticles. Besides, the silver plasmonic nanoparticles usually act as transducers that convert small changes in the local refractive index into spectral shifts in the intense nanoparticle extinction and scattering spectra, hence not only the different dielectric constant causes the LSPR spectral shift, but the complex adsorbents at the interface of the silver nanoparticles. We have added the above discussions into the revised manuscript. The exact relationship between the LSPR and the refractive of different liquids (including the biological liquids), as well as the application of the universal dispersibility of the Ag nanoparticles to detect the refractive index of different liquids will be studied further in the future.

Comment 3: Wet chemical methods are still among the most used methods of nanosilver preparation both for their simplicity and economic and time cost, but mainly for the ability to control the size and especially the shape and morphology of produced silver nanoparticles in contrast to limited controllable electrochemical preparation methods. Is it possible to use this method of nanosilver modification and the ability to disperse in polar and non-polar solvents also in the case of silver nanoparticles prepared by wet chemical reduction methods? If so, then it would greatly expand the possibilities of applying silver nanoparticles in practice. Given the ability to control the wet reduction process and purposefully control the size and shape of nanoparticles, this would be very beneficial and would move the synthesis

options forward, as well as application possibilities.

Response 3: We greatly appreciate for your constructive comments. According to your suggestion, we explored the possibility to modify the dodecyl sulfate ions onto the surface of the silver nanoparticles during wet chemical preparation. We first added SDS (10 mM) during the chemical reduction of silver nitrate with sodium borohydride (the most widely used wet chemical fabrication method). The prepared silver nanoparticles could disperse in water, while immediately aggregated in DMF and cyclohexane (Figure R2). This means that the dodecyl sulfate ions at least could not completely replace the adsorbed ions or ligands (such as the borohydride) during the preparation of silver nanoparticles by the wet chemical method. In other words, the dodecyl sulfate ions have weaker interactions to silver than the borohydride.

Fig. R2. Trying to bind SDS to the Ag nanoparticles during the wet chemical fabrication process. a to c, Photos of silver nanoparticles dispersed in water, and aggregated in DMF, and cyclohexane, respectively.

To further confirm the ligand status of the Ag nanoparticles, we added sodium borohydride (10 mM) to the colloidal solutions composed of the “smart” silver nanoparticles prepared by the electrochemical method. The dispersible properties of the obtained silver nanoparticles changed obviously (Figure R3a-c), exhibiting the same dispersible limit as the wet chemical method prepared ones: easily dispersed in water, but aggregated while met DMF and cyclohexane. We investigated the ligand composition of the obtained silver nanoparticles by XPS analysis, showing that both sulfur and boron elements existed on the surface of the obtained silver nanoparticles (Figure R3d and e). This means that borate ligands can partially replace the dodecyl sulfate ligands on the surface of the silver nanoparticles. In other words, the binding strength of the borate ligands to the silver nanoparticles should be higher than that of the dodecyl sulfate ligands. Therefore, it is possible to bind dodecyl sulfate ions to the Ag nanoparticles during the wet chemical preparation if a chemical reduction agent

(or its derivatives) that has weaker interactions to the Ag can be found. We will keep looking for the chemical reducing agent. We expect that we can bind the dodecyl sulfate ions to the wet chemically prepared Ag nanoparticles and make them dispersible in both polar and non-polar solvents. As you said, this will open many application possibilities.

Fig. R3. Surface ligand status of Ag nanoparticles. **a to c**, Photos of “smart” silver nanoparticles dispersed in water, and aggregated in DMF and cyclohexane, respectively, after introducing sodium borohydride into the solution. **d, e**, XPS spectra of silver nanoparticles obtained by adding sodium borohydride into the “smart” silver nanoparticle colloidal solutions.

REVIEWERS' COMMENTS

Reviewer #1 (Remarks to the Author):

The authors have carefully addressed the comments and clarified a number of important points in the manuscript. Overall, the revision has been carried out with care and I believe the manuscript will be of significant interest by the readership of Nature Communications. I recommend publication.

Reviewer #2 (Remarks to the Author):

The authors have successfully and sufficiently addressed all the comments, and therefore I recommend accepting this work for publication in Nature communication.

Responses to Referees' comments

Reviewer #1

Overall Comment: The authors have carefully addressed the comments and clarified a number of important points in the manuscript. Overall, the revision has been carried out with care and I believe the manuscript will be of significant interest by the readership of Nature Communications. I recommend publication.

Author Response: Thank you so much for recommending acceptance of our manuscript.

Reviewer #2

Overall Comment: The authors have successfully and sufficiently addressed all the comments, and therefore I recommend accepting this work for publication in Nature communication.

Author Response: Thank you so much for recommending acceptance of our manuscript.